# Feeding during Dialysis Increases Intradialytic Blood Pressure Variability and Reduces Dialysis Adequacy

**DOI:** 10.3390/nu14071357

**Published:** 2022-03-24

**Authors:** Elena Fotiadou, Panagiotis I. Georgianos, Vasilios Vaios, Vasiliki Sgouropoulou, Dimitrios Divanis, Apostolos Karligkiotis, Konstantinos Leivaditis, Michail Chourdakis, Pantelis E. Zebekakis, Vassilios Liakopoulos

**Affiliations:** 1Hemodialysis Unit, 1st Department of Medicine, AHEPA Hospital, Aristotle University of Thessaloniki, GR54636 Thessaloniki, Greece; elena_fotiadou@yahoo.gr (E.F.); pangeorgi@yahoo.gr (P.I.G.); vvaios_85@yahoo.gr (V.V.); vsgouro@hotmail.com (V.S.); ddivanis@gmail.com (D.D.); apokarli@gmail.com (A.K.); konleiv@windowsmail.com (K.L.); pzempeka@auth.gr (P.E.Z.); 2Laboratory of Hygiene, Social-Preventive Medicine and Biostatistics, School of Medicine, Aristotle University of Thessaloniki, GR54006 Thessaloniki, Greece; mhourd@gapps.auth.gr

**Keywords:** BP variability, hemodialysis, hypotension, intradialytic meals, urea reduction ratio

## Abstract

Whether hemodialysis patients should be allowed or even encouraged to eat during dialysis remains a controversial topic. This cross-over study aimed to evaluate the impact of feeding during dialysis on intradialytic blood pressure (BP) profile and dialysis adequacy in 26 patients receiving thrice-weekly, in-center hemodialysis. Over three consecutive mid-week dialysis sessions, intradialytic BP was monitored using the Mobil-O-Graph device (IEM, Stolberg, Germany). Blood samples were also obtained for the determination of the urea reduction ratio (URR). At baseline, patients underwent dialysis without the provision of a meal. In phases A and B, a meal with either high-protein (1.5 gr/kg of body weight) or low-protein (0.7 gr/kg of body weight) content was administered 1 h after the initiation of dialysis. The sequence of meals (high-protein and low-protein or vice versa) was randomized. Average intradialytic systolic BP (SBP) was similar on all three occasions. However, compared with baseline, the standard deviation (SD) (11.7 ± 4.1 vs. 15.6 ± 7.6 mmHg, *p* < 0.01), coefficient of variation (CV) (9.5 ± 3.7% vs. 12.4 ± 6.0%, *p* < 0.01) and average real variability (ARV) (9.4 ± 3.9 vs. 12.1 ± 5.2 mmHg, *p* < 0.01) of intradialytic SBP were higher in phase A. Similarly, compared with the baseline evaluation, all three indices of intradialytic SBP variability were higher in phase B (SD: 11.7 ± 4.1 vs. 14.1 ± 4.5 mmHg, *p* < 0.05; CV: 9.5 ± 3.7% vs. 11.1 ± 3.8%, *p* < 0.05; ARV: 9.4 ± 3.9 vs. 10.9 ± 3.9 mmHg, *p* < 0.05). Compared with dialysis without a meal, the consumption of a high-protein or low-protein meal resulted in a lower URR (73.4 ± 4.3% vs. 65.7 ± 10.7%, *p* < 0.001 in phase A and 73.4 ± 4.3% vs. 67.6 ± 4.3%, *p* < 0.001 in phase B, respectively). In conclusion, in the present study, feeding during dialysis was associated with higher intradialytic SBP variability and reduced adequacy of the delivered dialysis.

## 1. Introduction

Whether end-stage kidney disease (ESKD) patients on thrice-weekly, in-center hemodialysis should be allowed or encouraged to eat during the dialysis procedure is a topic that is surrounded by substantial controversy [1]. In such patients, comparative studies showed that caloric intake is lower during the dialysis-on than during the dialysis-off days [2], indicating that hemodialysis patients possibly skip three meals per week during their dialysis treatments. Accordingly, the administration of intradialytic meals may be a therapeutic opportunity to improve protein-energy wasting and health-related quality of life, particularly in malnourished hemodialysis patients [3,4]. However, these potential benefits to nutritional status may be counteracted by excess risk for symptomatic intradialytic hypotension and reduced dialysis adequacy [5]. 

The benefit/risk ratio of feeding during dialysis is not yet fully elucidated, taking into consideration that the currently available studies in this area have provided contradictory results. For example, some retrospective observational studies have shown that oral food intake during dialysis is not associated with an increased incidence of intradialytic hypotension [6,7]. These observational data contrast with the results of earlier interventional studies showing that the administration of intradialytic meals is accompanied by a more rapid postprandial decline in blood pressure (BP) and a more frequent occurrence of symptomatic hypotensive episodes during dialysis [5,8,9]. The controversy is further magnified by the fact that most of these interventional studies suffer from inherent methodological limitations, such as small sample sizes, lack of a randomized design, use of non-validated devices for intradialytic BP monitoring and substantial heterogeneity in the definition of intradialytic hemodynamic instability [5]. 

Therefore, the primary aim of this randomized, cross-over study was to investigate the impact on intradialytic BP profile and variability when feeding with either a high-protein or low-protein meal during dialysis. A secondary objective of our study was to explore whether the administration of intradialytic meals interferes with the adequacy of the delivered dialysis.

## 2. Materials and Methods

### 2.1. Study Population

Our study recruited ESKD patients receiving renal replacement therapy in the Hemodialysis Unit, 1st Department of Medicine, AHEPA University Hospital of Thessaloniki, Greece. Consecutive patients were screened for eligibility regardless of their diabetic status, antihypertensive drug use and a known history of intradialytic hypotension. The prespecified inclusion criteria were as follows: (i) age >18 years; (ii) in-center, thrice-weekly hemodialysis for at least 3 months prior to study enrollment; (iii) the patient must provide informed written consent. Patients were not eligible in the study in case of (i) inability for safe and independent oral food intake during dialysis; (ii) chronic atrial fibrillation; (iii) inability to obtain accurate BP recordings, such as in patients with a non-functioning arteriovenous fistula or graft in the contralateral arm from that currently used as vascular access for hemodialysis; (iv) concurrent infectious and/or bleeding complications; (v) hospitalization for acute myocardial infarction, unstable angina or stroke within 1 month prior to study enrollment; (vi) active malignant disease or other advanced comorbidities associated with poor life expectancy; (vii) unwillingness to sign the consent form.

The protocol procedures of our study were accordant with the Declaration of Helsinki and its latest amendments. All patients gave informed written consent before enrollment, and the protocol of our study was reviewed and approved by the ethics committee of the School of Medicine, Aristotle University of Thessaloniki (code of approval: 1.54/21-11-18). In addition, the study was registered in ClinicalTrials.gov with the identifier number NCT03947710.

### 2.2. Study Design

This study followed a randomized, cross-over design (Figure 1). In detail, at baseline (Week 0), patients were evaluated without oral food consumption during their scheduled dialysis treatments. Subsequently, patients were evaluated in 2 different study phases 1-week apart. In Phase A, a meal with a high-protein content (1.5 gr/kg of body weight) that included white bread and a grilled chicken breast or burger with mustard sauce, olive oil and cherry tomatoes was administered 1 h after the initiation of the 3 dialysis sessions of Week 1. In Phase B, a low-protein meal (0.7 gr/kg of body weight) containing white bread and margarine, jam, honey or cream cheese (according to the patient’s preference) was given 1 h after the start of the 3 dialysis sessions of Week 2. The sequence of intradialytic meals (high-protein and low-protein or vice versa) was random. Patients enrolled in this study were receiving renal replacement therapy with either hemodialysis or online hemodiafiltration. However, the mode of dialysis and all other parameters that may affect intradialytic hemodynamic response and/or dialysis adequacy (e.g., blood flow rate, dialysate flow rate, duration of dialysis session) were kept constant on all 3 occasions. Over the course of the study, patients were receiving their regular dialysis treatments, during which ultrafiltration volume was programmed according to their clinically set dry weight. In addition, the dry weight of study participants remained unmodified in order to reassure that study evaluations would be performed with similar ultrafiltration rates. 

### 2.3. Intradialytic BP Monitoring

Intradialytic BP was monitored in all 3 phases of the study during the mid-week dialysis session with the Mobil-O-Graph (IEM, Stolberg, Germany), an oscillometric device that incorporates a brachial BP detection unit that has been validated according to the criteria of the European Society of Hypertension [10]. Furthermore, comparative studies have shown that brachial BP recordings obtained with the Mobil-O-Graph device are comparable with BP measurements taken with other commercially available ambulatory BP monitors [11]. The Mobil-O-Graph device was fitted to the non-fistula (or non-dominant) arm using a cuff of appropriate size shortly before each mid-week dialysis session. Intradialytic BP was monitored at regular time-intervals (i.e., every 15 min). Subsequently, BP recordings were extracted from the Mobil-O-Graph software and entered into a purpose-built data-Excel sheet. For safety reasons, non-scheduled BP measurements were also allowed during dialysis but only when this was mandatory (i.e., in the case of a symptomatic hypotensive event). However, in an attempt to reassure the accurate evaluation of intradialytic BP profile, only measurements recorded at the prespecified time-points at which the device was programmed to record BP were utilized in this analysis. The median number of BP recordings obtained during dialysis was 16 (range: 11, 18). We calculated average intradialytic BP as well as the following indices of short-term BP variability using validated formulas: (i) standard deviation (SD); (ii) coefficient of variation (CV); (iii) average real variability (ARV) of intradialytic systolic and diastolic BP [12]. Intradialytic hypotension was defined according to the 2007 European Best Practice Guidelines (EBPG) on hemodynamic instability as a decrease in systolic BP ≥ 20 mmHg accompanied by clinical symptoms/events and the necessity for nursing interventions [13].

### 2.4. Dialysis Adequacy

Blood samples were acquired with the use of the slow flow method at the start and end of each mid-week dialysis session for the determination of plasma urea concentrations [14]. The urea reduction ratio (URR) was calculated with the following formula: (predialysis urea—postdialysis urea)/predialysis urea.

### 2.5. Statistical Analysis

Categorical variables were expressed as absolute frequencies and percentages (*n*, %). Continuous variables were presented as mean values ± SD or a median (range). The normality of the distribution was assessed with the Shapiro–Wilk test. Comparisons for continuous variables among the 3 phases of the study were performed with the paired Student’s t-test or the non-parametric Wilcoxon signed-rank test, according to the normality of the distribution of each variable. A *p* value < 0.05 (two-tailed) was considered as statistically significant. The analysis was conducted with the Statistical Package for Social Sciences, version 23 (SPSS Inc., Chicago, IL, USA).

## 3. Results

Of the 58 hemodialysis patients who were screened for eligibility, 18 patients did not fulfill the prespecified inclusion/exclusion criteria of our study. Of the 40 patients approached, 9 patients refused to participate, and another 5 patients were excluded from the analysis due to incomplete intradialytic BP monitoring in all three phases of the study. The baseline characteristics of study participants are depicted in Table 1. A total of 26 patients (20 males and 6 females) with a mean age of 60.5 ± 12.3 years and a median dialysis vintage of 44 months (range: 3, 272) participated in the study. These patients were receiving thrice-weekly, in-center hemodialysis with a median blood flow rate (Qb) of 300 mL/min (range: 250–350) and a median dialysate flow rate of 600 mL/min (range: 500–800). Diabetic nephropathy and chronic glomerulonephritis were the two most common primary causes of kidney failure. As expected, the burden of cardiovascular comorbidities in our cohort was high: 46.2% of patients had diabetes mellitus, 88.5% were hypertensives, 50% had dyslipidemia, 53.8% had a history of coronary heart disease and 11.5% had a history of congestive heart failure. The mean levels of body mass index and predialysis serum albumin were 25.3 ± 4.8 kg/m^2^ and 4.2 ± 0.3 g/dL, respectively. The use of antihypertensive medications was common: 46.2% of patients were being treated with β-blockers, 46.2% with long-acting dihydropyridine calcium-channel blockers, 57.7% with loop diuretics and 15.4% with agents blocking the renin-angiotensin system.

As shown in Table 2, Table 3 and Table 4, the mean ultrafiltration rate was 6.9 ± 3.7 mL/kg/h at baseline evaluation, 7.0 ± 3.3 mL/kg/h in the dialysis session with the high-protein meal and 6.4 ± 3.3 mL/kg/h in the dialysis session with the low-protein meal. The pair-wise comparisons showed that ultrafiltration rates did not significantly differ among the three mid-week dialysis sessions. Similarly, the mean levels of intradialytic systolic BP, diastolic BP and heart rate were similar on all three occasions. In contrast, as compared with baseline evaluation, the SD (11.7 ± 4.1 vs. 15.6 ± 7.6 mmHg, *p* < 0.01), CV (9.5 ± 3.7% vs. 12.4 ± 6.0%, *p* < 0.01) and ARV (9.4 ± 3.9 vs. 12.1 ± 5.2 mmHg, *p* < 0.01) of intradialytic systolic BP were significantly higher in phase A (Table 2). Similarly, as compared with the dialysis session without the provision of a meal, the consumption of a low-protein meal in phase B resulted in significantly higher levels of SD (11.7 ± 4.1 vs. 14.1 ± 4.5 mmHg, *p* < 0.05), CV (9.5 ± 3.7% vs. 11.1 ± 3.8%, *p* < 0.05) and ARV of intradialytic systolic BP (9.4 ± 3.9 vs. 10.9 ± 3.9 mmHg, *p* < 0.05) (Table 3). Intradialytic variability of diastolic BP was slightly higher with the consumption of high-protein and low-protein meals, but the levels of SD, CV and ARV of intradialytic diastolic BP in phases A and B did not significantly differ from the corresponding indices at the baseline evaluation (Table 2, Table 3 and Table 4). Symptomatic intradialytic hypotension requiring nursing interventions occurred in two patients at baseline evaluation, in three patients in phase A and in one patient in phase B.

With respect to the adequacy of the delivered dialysis, as compared with baseline, the URR was significantly lower in phase A (73.4 ± 4.3% vs. 65.7 ± 10.7%, *p* < 0.001) as well as in phase B (73.4 ± 4.3% vs. 67.6 ± 4.3%, *p* < 0.001). The levels of URR did not significantly differ between a dialysis session with a high-protein meal and a dialysis session with a low-protein meal (65.7 ± 10.7% vs. 67.6 ± 4.3%, *p* = 0.36) (Figure 2).

## 4. Discussion

Prior studies that aimed to explore the effect of feeding during dialysis on intradialytic hemodynamic stability suffered from inherent methodological limitations and provided discordant results [15]. In an observational study that retrospectively evaluated intradialytic BP recordings obtained over three consecutive dialysis sessions in 126 stable hemodialysis patients, the amount of oral food intake was not associated with the occurrence of intradialytic hypotension (defined as a systolic BP < 100 mmHg at any time-point during dialysis) [6]. In a 9-week, non-randomized, parallel-arm study, nine patients were given high-protein meals for 25 consecutive dialysis sessions, and another nine controls completed the study without oral food intake during their dialysis treatments. The administration of high-protein meals was not accompanied by a higher incidence of symptomatic intradialytic hypotensive events [7]. These observations contrast with the results of earlier interventional studies showing that the consumption of intradialytic meals provokes a more rapid postprandial reduction in BP, aggravating the risk for symptomatic intradialytic hypotension [8,9]. As an example, in a small cross-over study, 13 hemodialysis patients were evaluated in random order during two standard dialysis sessions (snack-HD and control-HD) that were conducted with similar ultrafiltration rates [9]. A significant drop in BP was observed during both dialysis sessions, but the rate of BP decline was significantly higher after the administration of a snack than during the corresponding period of dialysis performed without oral food intake. In addition, the frequency of symptomatic intradialytic hypotension requiring intravenous saline infusion was significantly higher during the snack-HD than during the control-HD [9]. 

The present work overcomes several of the limitations of prior interventional studies aiming to elucidate the controversial issue of intradialytic hemodynamic response to the administration of a high-protein or low-protein meal during dialysis. Unlike prior studies, patients in the present work were evaluated in three consecutive mid-week dialysis sessions following a randomized and cross-over design. Other strengths of the present study include the use of the validated Mobil-O-Graph device [16] that enabled more accurate intradialytic BP monitoring and the estimation of novel indices reflecting short-term BP variability. The main findings of the present study are as follows: (i) symptomatic intradialytic hypotensive events requiring nursing interventions rarely occurred, with an equal distribution among the three phases of the study; (ii) the average levels of intradialytic BP did not significantly differ between the dialysis sessions performed with and without the administration of intradialytic meals; (ii) compared with the dialysis session performed without the provision of a meal, the SD, CV and ARV of intradialytic systolic BP were consistently higher with the administration of either a high-protein or low-protein meal 1 h after the initiation of dialysis.

The finding of our study that oral food intake during dialysis is associated with increased short-term (i.e., intradialytic) BP variability is biologically plausible. Over the postprandial period, there is typically a redistribution of circulating blood volume arising from the increased blood supply to the splachnic circulation [17]. These alterations may subsequently affect the hemodynamic response of patients to ultrafiltration, thereby modifying their intradialytic BP profiles. The increased short-term BP variability with the consumption of intradialytic meals may also have prognostic implications if we take into consideration that observational studies have shown higher intradialytic BP variability to be associated with a higher risk for adverse cardiovascular events and mortality among patients on hemodialysis [18,19]. This risk association may be even stronger when increased BP variability is accompanied by a more frequent occurrence of symptomatic intradialytic hypotension, which is a serious and common dialysis-related complication that has also been associated with a heightened risk for cardiovascular morbidity and mortality [20,21]. In the present study, episodes of symptomatic intradialytic hypotension were infrequent, possibly due to the short-term course of the study. However, it has to be mentioned that our study was designed to investigate acute alterations in intradialytic BP profile over three consecutive mid-week dialysis sessions. Larger randomized clinical trials with careful intradialytic BP monitoring in multiple dialysis sessions are needed to confirm the consistency of our results over a longer follow-up period. 

A secondary objective of our study was to explore the effect of oral food intake on the adequacy of the delivered dialysis. As compared with the dialysis session performed without the provision of a meal, oral intake of a high-protein or low-protein meal resulted in a lower URR in phases A and B of our study. The minimal residual renal function in patients with diuresis > 500 ml/24 H does not influence the blood pressure outcome during intra-dialytic session. Once again, the mechanistic substrate for this unfavorable effect is not fully clear, but increased blood pooling in the splachnic circulation during the postprandial period and the subsequent redistribution in circulating blood volume may be a plausible explanation [17]. This finding is in accordance with the results of two prior interventional studies showing that feeding during dialysis interferes with dialysis adequacy [22,23]. In the first study, 14 stable hemodialysis patients were evaluated during two consecutive mid-week dialysis sessions performed with identical blood flow and dialysate flow rates. The URR (71.5 ± 5.9% vs. 73.5 ± 6.6%, *p* < 0.05) and single-pool Kt/V (1.54 vs. 1.65, *p* < 0.05) were lower during the fed than during the fasting dialysis session [23]. In a subsequent study with a similar design, 25 non-diabetic hemodialysis patients were evaluated in two consecutive mid-week sessions (1-week apart) performed with and without the consumption of a standard meal 1 h after the initiation of dialysis. Once again, oral food intake during dialysis led to a significantly lower URR (67.8 ± 6.1% vs. 72.1 ± 6.0%, *p* < 0.001) and single-pool Kt/V (1.4 ± 0.2 vs. 1.6 ± 0.2, *p* <0.001) [22]. 

## 5. Conclusions

In conclusion, the present study shows that among ESKD patients receiving thrice-weekly, in-center hemodialysis, the administration of a standard intradialytic meal with high-protein or low-protein content was associated with a significantly higher intradialytic BP variability and a lower clearance of urea during dialysis. Whether these risks are counteracted by a beneficial effect of intradialytic meals on nutritional status that may favorably affect long-term clinical outcomes remains a crucial research question that warrants further investigation in large randomized controlled trials in the future. 

## Figures and Tables

**Figure 1 nutrients-14-01357-f001:**
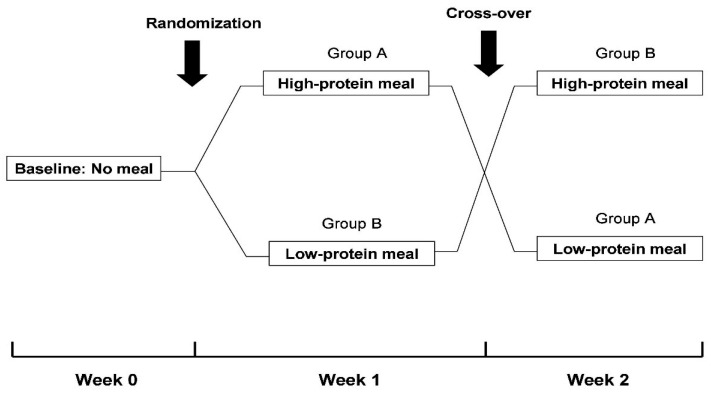
Flowchart of study design.

**Figure 2 nutrients-14-01357-f002:**
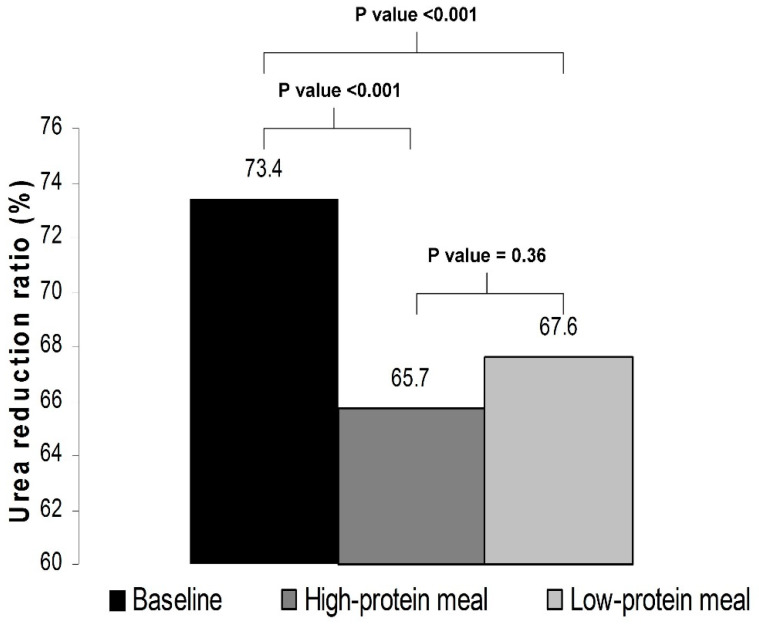
Urea reduction ratio at baseline and in phases A and B of the study.

**Table 1 nutrients-14-01357-t001:** Clinical and laboratory characteristics of study participants.

*N*	26
Demographics	
Age (years)	60.5 ± 12.3
Male gender (*n*, %)	20, (76.9%)
BMI (kg/m^2^)	25.3 ± 4.8
Primary cause of ESKD (*n*, %)	
Diabetic nephropathy	9, (34.6%)
Hypertensive nephrosclerosis	4, (15.4%)
Glomerulonephritis	6, (23.1%)
Other	3, (11.5%)
Unknown	4, (15.4%)
Comorbidities (*n*, %)	
Diabetes	12, (46.2%)
Hypertension	23, (88.5%)
Dyslipidemia	13, (50.0%%)
History of CHD	14, (53.8%)
History of CHF	3, (11.5%)
Dialysis parameters	
Dialysis vintage (months)	44 (3, 272)
Mode of dialysis (*n*, %)	
*HD*	17, (65.4%)
*On-line HDF*	9, (34.6%)
Vascular access	
*Arteriovenous fistula*	14, (53.8%)
*Central venous catheter*	12, (46.2%
Blood flow―Qb (mL/min)	300, (250–350)
Dialysate flow―Qd (mL/min)	600, (500–800)
Residual diuresis ≥0.5 L/24-h (*n*, %)	15, (57.7%)
Laboratory parameters	
Hemoglobin (g/dl)	11.5 ± 1.0
Urea (mg/dL)	140.8 ± 35.1
Urea reduction ratio (%)	73.4 ± 4.3
Creatinine (mg/dl)	8.1 ± 1.9
Calcium (mg/dL)	8.8 ± 0.7
Phosphate (mg/dL)	5.3 ± 1.4
Albumin (g/dL)	4.2 ± 0.3
Antihypertensive medications (*n*, %)	
β-blocker	12, (46.2%)
ACEIs or ARBs	4, (15.4%)
CCBs	12, (46.2%)
Loop diuretics	15, (57.7%)
Centrally-acting agents	2, (7.7%)

Abbreviations: ACEI = angiotensin-converting-enzyme-inhibitor; ARB = angiotensin-receptor-blocker; CCB = calcium-channel-blocker; CHD = coronary heart disease; CHF = congestive heart failure; HD = hemodialysis; HDF = hemodiafiltration; MI = myocardial infarction.

**Table 2 nutrients-14-01357-t002:** Intradialytic BP variability parameters at baseline and in phase A of the study.

Parameter	Baseline	High-Protein Meal	*p* Value
Average intradialytic SBP (mmHg)	124.6 ± 17.2	127.1 ± 17.0	0.38
Intradialytic SBP-SD (mmHg)	11.7 ± 4.1	15.6 ± 7.6	<0.01
Intradialytic SBP-CV (%)	9.5 ± 3.7	12.4 ± 6.0	<0.01
Intradialytic SBP-ARV (mmHg)	9.4 ± 3.9	12.1 ± 5.2	<0.01
Average intradialytic DBP (mmHg)	78.9 ± 10.2	79.0 ± 11.4	0.98
Intradialytic DBP-SD (mmHg)	8.3 ± 2.6	9.7 ± 0.9	0.17
Intradialytic DBP-CV (%)	10.6 ± 3.5	12.5 ± 5.7	0.14
Intradialytic DBP-ARV (mmHg)	7.2 ± 2.1	7.9 ± 3.1	0.41
Average intradialytic HR (bpm)	70.7 ± 12.1	72.9 ± 10.8	0.16
Ultrafiltration volume (L)	1.9 ± 0.9	2.0 ± 0.8	0.79
Ultrafiltration rate (mL/kg/h)	6.9 ± 3.7	7.0 ± 3.3	0.97

Abbreviations: ARV = average real variability; CV = coefficient of variation; DBP = diastolic blood pressure; SBP = systolic blood pressure; SD = standard deviation.

**Table 3 nutrients-14-01357-t003:** Intradialytic BP variability parameters at baseline and in phase B of the study.

Parameter	Baseline	Low-Protein Meal	*p* Value
Average intradialytic SBP (mmHg)	124.6 ± 17.2	129.9 ± 18.3	0.11
Intradialytic SBP-SD (mmHg)	11.7 ± 4.1	14.1 ± 4.5	<0.05
Intradialytic SBP-CV (%)	9.5 ± 3.7	11.1 ± 3.8	<0.05
Intradialytic SBP-ARV (mmHg)	9.4 ± 3.9	10.9 ± 3.9	<0.05
Average intradialytic DBP (mmHg)	78.9 ± 10.2	80.7 ± 12.9	0.37
Intradialytic DBP-SD (mmHg)	8.3 ± 2.6	8.6 ± 2.9	0.66
Intradialytic DBP-CV (%)	10.6 ± 3.5	11.0 ± 3.8	0.72
Intradialytic DBP-ARV (mmHg)	7.2 ± 2.1	7.3 ± 2.2	0.92
Average intradialytic HR (bpm)	70.7 ± 12.1	72.2 ± 11.0	0.21
Ultrafiltration volume (L)	1.9 ± 0.9	1.8 ± 1.0	0.61
Ultrafiltration rate (mL/kg/h)	6.9 ± 3.7	6.4 ± 3.3	0.23

Abbreviations: ARV = average real variability; CV = coefficient of variation; DBP = diastolic blood pressure; SBP = systolic blood pressure; SD = standard deviation.

**Table 4 nutrients-14-01357-t004:** Comparison of intradialytic BP variability parameters between phases A and B of the study.

Parameter	High-Protein Meal	Low-Protein Meal	*p* Value
Average intradialytic SBP (mmHg)	127.1 ± 17.0	129.9 ± 18.3	0.24
Intradialytic SBP-SD (mmHg)	15.6 ± 7.6	14.1 ± 4.5	0.32
Intradialytic SBP-CV (%)	12.4 ± 6.0	11.1 ± 3.8	0.26
Intradialytic SBP-ARV (mmHg)	12.1 ± 5.2	10.9 ± 3.9	0.23
Average intradialytic DBP (mmHg)	79.0 ± 11.4	80.7 ± 12.9	0.26
Intradialytic DBP-SD (mmHg)	9.7 ± 0.9	8.6 ± 2.9	0.15
Intradialytic DBP-CV (%)	12.5 ± 5.7	11.0 ± 3.8	0.12
Intradialytic DBP-ARV (mmHg)	7.9 ± 3.1	7.3 ± 2.2	0.31
Average intradialytic HR (bpm)	72.9 ± 10.8	72.2 ± 11.0	0.60
Ultrafiltration volume (L)	2.0 ± 0.8	1.8 ± 1.0	0.44
Ultrafiltration rate (mL/kg/h)	7.0 ± 3.3	6.4 ± 3.3	0.18

Abbreviations: ARV = average real variability; CV = coefficient of variation; DBP = diastolic blood pressure; SBP = systolic blood pressure; SD = standard deviation.

## Data Availability

The patient data used for this study are restricted and not publicly available due to privacy and ethical concerns. Please contact the corresponding author, as these data may potentially be shareable with appropriate permissions and oversight.

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
