# Peer review of "Feeding during Dialysis Increases Intradialytic Blood Pressure Variability and Reduces Dialysis Adequacy"

_nutrients, 2022, doi:10.3390/nu14071357_

Round 1
Reviewer 1 Report
The authors have examined the influence of meals during dialysis on intradialytic BP (and URR). The number of patients isn't large, although the cross-over design helps with this.
With regard to the design - it may have been better to randomly assign the 3 periods (no-meal, low-protein, high-protein). I would also have preferred to see each test period include meals with each dialysis session for at least a week prior to testing, to ensure there was no cross-over effect.
Given that this was primarily a BP study - it would be useful to detail the ultrafiltration rate (in mls/kg/hr, as is customary) for each test period.
I've never been overly convinced about the usefulness of BP variability and its association with outcome. One is always concerned about this in a single dialysis session per patient, per test condition. I do wonder whether this would remain significant if multiple sessions were examined with each patient and test condition.
The authors don't offer an explanation for the URR reduction - I think this should be discussed in the Discussion.
Reviewer 2 Report
Fotiadou et al. evaluated the intradialytic blood pressure profile and dialysis adequacy in hemodialysis patients. The researcher found feeding during dialysis was associated with higher intradialytic SBP variability and reduced adequacy of the delivered dialysis. The study is interesting, but small sample sizes and a lack of a randomized design are major drawbacks. I have some specific comments for this study.
- A study flowchart is suggested to demonstrate the study design.
- As for the baseline characteristics of study participants, the residual renal function, residual daily urine amount, and general dialysis clearance (such as kt/v) could be demonstrated.
- In Table 2, the results demonstrated differences in the intradialytic SBP variability parameters but not intradialytic DBP variability parameters. A brief discussion is suggested in the discussion section.
- How many time points were used to calculate the BP variability (SD, CV, ARV)? This should be particularly detailed described in the method section.
Reviewer 3 Report
Authors present a randomized, cross-over study about the impact of feeding during dialysis on intradialytic blood pressure and dialysis adequacy in end stage kidney disease patients receiving thrice-weekly hemodialysis. They affirmed that the administration of a standard intradialytic meal with high-protein or low-protein content was associated with a significantly higher intradialytic blood pressure variability and with a lower clearance of urea during dialysis.
This is a good and interesting article. Some minor revisions are needed before the publication of this article. The iconographic support must necessarily be implemented and better organized.
Firstly, we suggest making the title more contracted and less didactic.
The abstract is clear. However, the authors should report data on standard deviation, coefficient of variation and average real variability of intradialytic systolic blood pressure of phase B from baseline, just as they do for phase A.
Materials and methods could be improved:
-In section “2.1 Study population”, it should be specified that patients with diabetes or/and on antihypertensive therapy were admitted to the study as mentioned in the results. It is also unclear whether patients with a personal history of intradialytic hypotension were admitted.
-In section “2.2 study design”, it should be mentioned that two different dialysis techniques (HD and HDF) were used on study population. This point determines inhomogeneity in the analyzed population, especially in relation to the higher level of hemodynamic stability that HDF is able to guarantee, according to the most recent literature. Please specify both in this section and in the discussion if possible.
- In section "2.3 Intradialytic BP monitoring", authors affirmed that "Intradialytic BP was monitored in all 3 phases of the study with the Mobil-O-Graph". However, a few lines further they state that "Manual BP measurements were also allowed". It is therefore not clear the number of manual measurements compared to those via Mobile-O-Graph. It could affect the accuracy of the measurements, particularly because authors defined the use of Mobil-O-Graph as a “strength point”.
Results and Discussions are clear. I suggest breaking down “Table 2” into three separate tables: the first to compare the baseline with phase A, the second to compare the baseline with phase B and the last one to compare phase A and phase B. Perhaps it might be useful to add more charts like Figure 1, using the data in Table 2.
